# Relationship between Preoperative Pyuria and Bacille Calmette-Guerin Treatment in Intravesical Recurrence after Transurethral Resection of High-Risk, Non-Muscle Invasive, Bladder Carcinoma: A Retrospective Study of Human Data

**DOI:** 10.3390/cancers15061638

**Published:** 2023-03-07

**Authors:** Ryo Tsukamoto, Tomokazu Sazuka, Yoshinori Hattori, Hiroaki Sato, Takayuki Arai, Yusuke Goto, Yusuke Imamura, Shinichi Sakamoto, Tomohiko Ichikawa

**Affiliations:** Department of Urology, Chiba University Graduate School of Medicine, 1-8-1 Inohana, Chuo-ku, Chiba 260-8670, Japan

**Keywords:** BCG, intravesical recurrence, non-muscle invasive bladder cancer, pyuria, risk factor

## Abstract

**Simple Summary:**

Some researchers have found that preoperative pyuria is a risk factor for the recurrence of high-risk non-muscle invasive bladder cancer. However, no one has determined the risks of pyuria in relation to bacille Calmette-Guerin (BCG) treatment status. We defined pyuria as ≥10 white blood cells per high-power field. We analyzed recurrence-free rates in 424 patients with high-risk non-muscle invasive bladder cancer who had and had not undergone BCG treatment. We identified no risk factors for recurrence in the BCG-treated group (*n* = 179). However, in the non-BCG treated group (*n* = 245), pyuria was associated with being female and having T1 cancer. According to univariate and multivariate analysis, preoperative pyuria is an independent risk factor for intravesical recurrence. No significant difference in the severity of pyuria was found between BCG and non-BCG-treated groups. Aggressive BCG treatment may need to be considered in patients with high-risk non-muscle invasive bladder cancer and pyuria.

**Abstract:**

Some researchers have found that preoperative pyuria is a risk factor for recurrence after transurethral resection of high-risk non-muscle invasive bladder cancer. However, to our knowledge, none have clarified the risks associated with pyuria according to bacille Calmette-Guerin (BCG) treatment status. We retrospectively selected patients with high-risk non-muscle invasive bladder cancer according to Japanese Urological Association guidelines. Pyuria was defined as ≥10 white blood cells per high-powered field. We analyzed recurrence-free rates (RFS) in 424 patients who had and had not undergone BCG treatment. The median duration of follow-up was 45.2 months. According to multivariate analysis, postoperative intravesical BCG induction and preoperative pyuria were independent risk factors for intravesical recurrence in the whole study cohort. We found no significant risk factors for recurrence in the BCG-treated group (*n* = 179). In the non-BCG-treated group (*n* = 245), patients with pyuria were much more frequently female and more often had T1 disease than patients without pyuria. According to univariate and multivariate analysis, preoperative pyuria is an independent risk factor for intravesical recurrence. There was no significant difference in the severity of pyuria between the BCG and non-BCG-treated groups. Aggressive BCG treatment may need to be considered in patients with high-risk NMIBC and pyuria.

## 1. Introduction

Bladder cancer is the 12th most common cancer worldwide, accounting for 573,278 new cases and 212,536 deaths annually [1]. Newly diagnosed bladder cancers are non-muscle invasive bladder cancers (NMIBCs) in 70% to 75% of patients [2,3]. The rate of recurrence of bladder carcinoma after transurethral resection (TUR) ranges from approximately 15% to 70% according to the European Organisation for Research and Treatment of Cancer recurrence scores. Various clinical and pathological risk factors for recurrence have been investigated [4]. These factors are routinely collected and are very important for risk classification; however, the identification of more risk factors is necessary. The recurrence rate of NMIBC is not low and recurrences are characteristically invasive and have adverse economic consequences [2,3,5]. Suppression of recurrence requires appropriately combining perioperative intravesical instillation therapy and TUR of the bladder tumor (TUR-Bt).

BCG therapy is widely used worldwide for NMIBC patients. There are also slight differences in indications for each guideline. A previous report that analyzed 1228 patients with non-muscle invasive bladder cancer showed multifocality, lymphovascular invasion, and high-grade re-TURB were independent predictors for response to BCG treatment. BCG-unresponsive patients reported worse oncological outcomes [6]. As mentioned above, it is true that there are some factors that are less effective for BCG treatment. There is also a report that inflammatory markers also predict the effect of BCG treatment. The modified Glasgow prognostic score has attracted attention in recent years. That might have the potential to predict recurrence in 1382 high-grade/Grade 3 NMIBC patients [7]. In this way, various studies on BCG are currently underway, and there are many issues that must be resolved.

The findings of our previous study of NMIBC suggested that preoperative pyuria contributes to the recurrence of bladder cancer. It has been postulated that pyuria promotes intravesical recurrence because the cancer cells temporarily activated by TUR are more likely to engraft on damaged mucous membranes within the bladder [8]. We have also reported that preoperative pyuria is a risk factor for intravesical recurrence after nephroureterectomy for upper urinary tract urothelial carcinomas [9]. In our previous study of NMIBC, we analyzed a wide range of high- to low-risk NMIBCs. In that study, high-risk patients underwent BCG treatment, whereas low-risk patients only underwent TUR-Bt. Because additional treatment of different intensity is administered according to NMIBC risk after TUR-Bt for this condition, caution is required when interpreting postoperative recurrence rates. Relapse after intensive BCG treatment differs from relapse without additional treatment after TUR-Bt. Intravesical instillation of BCG is highly effective in reducing the risk of intravesical recurrence and progression of NMIBCs [10,11]. Comparing cohorts with significantly different risks of intravesical recurrence is problematic. Past retrospective studies have reported that pyuria is a predictor of poor prognosis in patients who have undergone BCG treatment [12]. However, to the best of our knowledge, no studies have determined the risks of pyuria according to BCG treatment status. Analyzing the relationship between preoperative pyuria and BCG treatment requires the inclusion only of high-risk patients, for which BCG treatment is the standard treatment after TUR-Bt. In Japan, BCG treatment is recommended for patients classified as at high risk according to the 2019 version of the Japanese Urological Association (JUA) bladder cancer guideline [13]. However, in some cases, BCG treatment is not instituted for various reasons. For example, some patients refuse BCG treatment because of the associated frequent urination, or because they are receiving immunosuppressive therapy. In the present study, we retrospectively examined real-world cases in a single institution and evaluated the risk of preoperative pyuria associated with intravesical recurrence according to BCG treatment status.

## 2. Materials and Methods

In the present study, we analyzed the data of patients who had undergone TUR-Bt for NMIBC between November 2008 and March 2022. Our focus was intravesical recurrence after TUR-Bt. This study was approved by the institutional review board of Chiba University (No. 2554).

We classified risk according to the JUA guideline for NMIBC [13]. We excluded patients with pathologically identified muscle-invasive bladder carcinoma, at low or intermediate risk according to the JUA guideline, and with other than bladder carcinoma. High risk is defined by the JUA as T1 or carcinoma in situ (CIS) or high grade.

In our hospital, preventive intravesical therapy with an anticancer agent is administered postoperatively to all patients, unless it is omitted at the surgeon’s discretion. Second, TUR is performed on all patients with T1 and/or high-grade disease. Intravesical BCG induction therapy is instituted in intermediate- and high-risk patients, and maintenance BCG intravesical therapy is for high-risk patients with two of the three risk factors of T1, CIS, or high grade.

The following data were retrospectively collected from hospital records: age, sex, preoperative pyuria, preoperative urine cytology, history of NMIBC, tumor grade, pathological T stage, concomitant CIS, and postoperative intravesical BCG induction. BCG was defined as having been administered even if it had only been administered once as induction or maintenance. Pyuria was defined as ≥10 white blood cells (WBCs) per high-power field (hpf) prior to TUR-Bt. Antibiotics were not administered preoperatively to patients with asymptomatic pyuria, but only to those with symptomatic urinary tract infections. On the day of TUR-Bt, all patients, with and without preoperative pyuria, received one dose of intravenous antibiotics. Positive preoperative urine cytology was defined as “positive” or class4 class5. Recurrence was defined as the diagnosis of intravesical bladder carcinoma or metastasis. Intravesical recurrence was diagnosed by cystoscopy, transurethral resection, urine cytology, or computed tomography.

All statistical analyses were carried out using JMP Pro 16.0.1 (SAS Institute, Cary, NC, USA). A two-sided *p*-value of <0.05 was considered to denote statistical significance. Comparisons were carried out using the Mann–Whitney and χ^2^ tests. Risk factors related to intravesical recurrence were analyzed by the Cox proportional hazards model and Kaplan–Meier method. When the *p*-value was 0.2 or less in univariate analysis, the factor was subjected to multivariate analysis.

## 3. Results

During the study period, we treated 458 patients, 34 of whom were excluded because they were at JUA low or intermediate risk, leaving 424 eligible patients. BCG treatment had been administered to 179 of these patients and not to the remaining 245. Forty-five of those who received BCG had preoperative pyuria and 134 did not. Thirty-eight of those who did not receive BCG had preoperative pyuria and 245 did not (Figure 1).

The median duration of follow-up of all study 424 patients was 45.2 months. During the study period, 115 events (bladder recurrence) occurred. Table 1 shows the patients’ backgrounds. The median age was 72.5 years. Those with pyuria were older than those without it (*p* = 0.041). Patients with pyuria were more often female than those without it (*p* = 0.019). BCG treatment was administered more often to those with pyuria than to those without it (*p* = 0.0136). Pathological T1 was found more often in those with pyuria than in those without it (*p* = 0.0028).

We examined the risk factors for postoperative bladder recurrence in all 424 patients (Table 2). According to univariate and multivariate analysis, postoperative intravesical BCG induction (hazard ratio [HR] 0.52, presence vs. absence, *p* = 0.0026) and preoperative pyuria (HR 1.93, presence vs. absence, *p* = 0.0329) were independent risk factors for intravesical recurrence. RFS according to pyuria status in all cases is shown in Figure 2 (HR 1.42; 95% CI, 0.92–2.21, presence vs. absence, *p* = 0.1125 according to univariate analysis).

As stated in Table 2, we divided the patients into two groups (BCG-treated vs. not BCG-treated) to determine whether BCG treatment is a strong independent risk factor for RFS. In the BCG-treated group (*n* = 179), patient baseline characteristics did not differ significantly between those with and without pyuria (Appendix A). We found no significant differences in RFS risk factors, including preoperative pyuria, between the BCG-treated and not BCG-treated groups (Table 3). Figure 3 shows Kaplan–Meier RFS curves according to pyuria status in BCG-treated patients (HR 1.50; 95% CI, 0.69–3.27, presence vs. absence, *p* = 0.2985).

In the non-BCG-treated group (*n* = 245), a greater proportion of female than male patients had pyuria and there was a greater proportion of T1 tumors in those without pyuria (Appendix A). According to univariate and multivariate analysis, the presence of preoperative pyuria (HR 1.77; 95% CI, 1.03–3.04; *p* = 0.0371) and positive preoperative cytology (HR 1.84; 95% CI, 01.01–3.33; *p* = 0.0459) were independent risk factors for intravesical recurrence (Table 4). Figure 4 shows Kaplan–Meier curves according to pyuria status in non-BCG-treated patients (HR 1.70; 95% CI, 1.00–2.91, presence vs. absence, *p* = 0.0485). The findings differed from those of BCG-treated patients.

We analyzed RFS according to the risk factors of pyuria and positive cytology as follows: low risk: cytology (−) and pyuria (−); intermediate risk: cytology (+) and pyuria (−)/cytology (−) and pyuria (+); high risk: cytology (+) and pyuria (+). Because there were only three patients in the high-risk group, we could not analyze them statistically. There was a significant difference between the low- and intermediate-risk groups (*p* = 0.0064) (Appendix A).

We evaluated RFS with and without BCG treatment according to the severity of pyuria, which was classified into the following three groups: urine WBCs 10–20/hpf, 20–100/hpf, and >100/hpf. In the BCG-treated group, patients with urine WBCs 10–20/hpf tended to have a slightly better RFS; however, this difference was not significant (Figure 5a). There were also no significant differences in RFS according to the severity of pyuria in the non-BCG group (Figure 5b). Thus, in this study, the severity of pyuria had no significant association with RFS.

In this study, preoperative pyuria was a significant prognostic factor for RFS only in non-BCG-treated patients.

## 4. Discussion

When treating NMIBC, it is important to suppress progression and recurrence. In particular, the intravesical recurrence rate is not low. When recurrence occurs, it creates a physical and economic burden for the patient [2,3,5,14,15]. Numerous studies have investigated risk factors for the intravesical recurrence of NMIBC. Some clinical and pathological factors have been shown to help in predicting the risk of recurrence. The main established predictors of recurrence are multiple tumors, tumor size, prior recurrence, tumor stage, and histological grade. These factors are used to classify risk in all guidelines, including the JUA guideline [13,16,17,18,19]. Adjuvant therapy, including perioperative intravesical instillation, can be administered in addition to TUR-Bt when treating NMIBC. It is important to identify patients at high risk and provide appropriate adjuvant treatment.

Each guideline now defines risk and determines additional treatment accordingly. The representative classifications and treatment methods are shown below by risk. Currently, the highest risk is also defined, but here we show low, intermediate, and high risk, respectively. By the AUA/SUO risk stratification, low-risk NMIBC includes papillary urothelial neoplasm of low malignant potential and low-grade urothelial carcinoma that is a solitary Ta and <3 cm. A single instillation of intravesical chemotherapy immediately post-TUR-Bt can be helpful in reducing the risk of recurrence. Intermediate-risk NMIBC includes low-grade urothelial carcinoma that has any of the following characteristics: T1, size >3 cm, multifocal, or recurrence within 1 year. In addition, high-grade urothelial carcinoma that is solitary, Ta, and <3 cm is also considered intermediate risk. After TUR-Bt and immediate intravesical chemotherapy, the panel recommends a 6-week induction course of intravesical therapy. Options for intravesical therapy for intermediate-risk NMIBC include BCG or chemotherapy. High-risk NMIBC includes high-grade urothelial carcinoma that has any of the following characteristics: CIS, T1, size >3 cm, or multifocal. Treatment options for high-risk NMIBC depend on whether the tumor has previously been shown to be unresponsive or intolerant to BCG. For BCG-naïve NMIBC, the options are cystectomy or BCG [20]. By the EAU risk stratification, low-risk NMIBC includes a primary, single, Ta/T1 LG/G1 tumor <3 cm in diameter without CIS in a patient aged <70 years. “Offer one immediate instillation of intravesical chemotherapy after TUR-Bt” is strongly recommended. Intermediate-risk NMIBC includes patients without CIS who are not included in either the low, high, or very high-risk group. “For all patients, either one year full-dose BCG treatment (induction plus 3-weekly instillations at 3, 6, and 12 month) or instillations of chemotherapy (the optimal schedule is not known) for a maximum of one year is recommended. The final choice should reflect the individual patient’s risk of recurrence and progression as well as the efficacy and side effects of each treatment modality. Offer one immediate chemotherapy instillation to patients with small papillary recurrences detected more than one year after previous TUR-Bt” is strongly recommended. High-risk NMIBC includes all T1 HG/G3 without CIS except those included in the very high-risk group, all CIS patients except those included in the very high-risk group, Ta LG/G2 or T1 G1 with CIS and all three risk factors, Ta HG/G3 or T1 LG with no CIS and at least two risk factors and T1 G2 with no CIS and at least one risk factor. (Additional risk factors: age >70 years, multiple papillary tumors, and tumor diameter >3 cm.) “Offer intravesical full-dose BCG instillations for 1–3 year or RC” is strongly recommended [21]. By the JUA risk stratification, low-risk NMIBC includes initial diagnosis, <3 cm, Ta, low grade, without CIS. Low-risk patients receive a single immediate intravesical instillation of anthracyclines or mitomycin C to prevent a recurrence. Intermediate-risk NMIBC includes the group meets other than low and high risk. Administration of maintenance intravesical chemotherapy in addition to single immediate intravesical instillation of chemotherapy has been recommended for patients with intermediate-risk disease. However, no conclusions about the regimen have been reached. In addition, intravesical BCG therapy is administered to intermediate risk. High-risk NMIBC includes the group containing any of the following factors: T1, high grade, CIS (including concurrent CIS). Intravesical BCG therapy is administered to high-risk patients. The addition of maintenance therapy after six to eight courses of intravesical induction therapy is recommended [13].

Recent study results suggest an association between bladder cancer incidence and several food items including meat, fruit, vegetables, milk products, and oil. Micronutrient deficiency is associated with bladder cancer risk. It is necessary to further examine the effect of such eating habits on relapse in the future [22].

We have previously shown that the presence of residual urine and pyuria are risk factors for intravesical recurrence after TUR-Bt in a wide range of patients with NMIBC [8]. There is insufficient robust evidence concerning pyuria and intravesical recurrence. Pyuria is easy to identify in daily practice and this investigation has economic advantages. Simplicity and low cost are desirable characteristics of biomarkers. In a previous study, we found that damaged mucosa and a large amount of residual urine are associated with an increased risk of intravesical recurrence because these factors result in many cancer cells remaining in the bladder after TUR-Bt. However, many patients with high-risk NMIBC undergo intravesical BCG instillation. Because BCG treatment is considered highly effective, the risk of recurrence is strongly dependent on whether or not this treatment is administered [23,24]. BCG treatment must be carefully considered when discussing postoperative recurrence of NMIBC with reference to guidelines. Various detailed analyses of BCG treatment are underway [25,26,27,28].

In the present study, we evaluated the relationship between preoperative pyuria and intravesical recurrence after TUR-Bt according to postoperative BCG induction therapy status using real-world single-institution experience. We showed that preoperative pyuria is not associated with intravesical recurrence after TUR-Bt in high-risk BCG-treated patients with NMIBC, but is a significant risk factor in non-BCG-treated patients.

First, the presence of pyuria is associated with more serious bladder cancer, including the depth of invasion, than the absence of pyuria is. We believe that pyuria occurs more frequently in patients with large tumors and extensive mucosal lesions, including CIS [29,30,31,32]. Most malignant tumors have significant leucocytic infiltrates and these have been linked to a poor prognosis [33,34]. In colorectal cancer, abundant tumor budding, scanty lymphocyte infiltration, and few lymphoid follicles tend to coexist and appear to be reliable prognostic indicators [35]. Leucocyte infiltrates are attracted to tumors by chemotactic factors released by viable or necrotic tumor cells or by cells within the tumor stroma [36]. Angiogenesis is a hallmark of malignant neoplasia because the formation of new blood vessels is required for tumors to access sufficient oxygen and nutrients for their continued growth and metastasis [37]. There is increasing evidence for communication and interactions between polymorphonuclear neutrophils and T cells in various physiological and pathological conditions, including acute and chronic inflammatory disease and defense against tumors [38]. Regulatory T cells throughout a tumor, including in tertiary lymphoid structures, are associated with poor outcomes in patients with non-small cell lung cancers [39]. Thus, there are numerous basic and clinical studies on the relationship between inflammatory cell infiltration and cancer. Accordingly, we believe that pyuria is a significant risk factor for recurrence in patients who have not undergone BCG treatment in association with TUR-Bt. We believe that TUR damages the mucous membrane within the bladder, temporarily enhancing the ability of cancer cells to engraft there [8].

Conversely, in this study preoperative pyuria was not a significant predictor of recurrence in patients who had undergone intravesical BCG instillation. However, other studies have found that the presence of pyuria is a risk factor for RFS in patients who have undergone BCG treatment [29,32]. Our results are contradictory, suggesting as they do that BCG treatment has a strong impact on the risk of intravesical recurrence, even in patients with pyuria. Patients without pyuria may have a lower risk of recurrence; however, BCG treatment can still be expected to suppress recurrence in the presence of pyuria. The superiority of any BCG strain over another could not be demonstrated yet. A previous report showed that when routinely performing re-TUR followed by a maintenance BCG schedule, TICE was superior to RIVM for RFS outcomes. Further studies addressing the type of BCG may also be needed in the future [40].

It may be necessary to reconsider the definition of pyuria. Various definitions of urinary tract infections have been used over the years [41,42,43]. In this study, we defined pyuria as 10 or more WBCs/hpf. Using a different definition may alter the results of the research. However, among the patients with pyuria in our study cohort, we found no differences in the association between the severity of pyuria and RFS between the BCG and non-BCG groups. The usability of an index is reduced if it cannot easily be assessed in daily clinical practice. The relatively small number of patients in this study made it difficult to obtain definitive results. We did not investigate bacteriuria in this study. Further large studies are needed to analyze the impact of the degree of pyuria and bacteriuria.

Based on previous studies and the results of this study, we believe that treatment for lower urinary tract symptoms in men may reduce the risk of NMIBC recurrence. Oral treatments such as alpha-blockers and 5-alpha-reductase inhibition, as well as surgical treatment, are expected to improve dysuria, thereby reducing residual urine volume, and improving chronic urinary tract infections [44]. In the future, our goal is to show that lower urinary tract obstruction treatment before TUR-Bt leads to a lower recurrence rate of NMIBC.

Our results indicate that pyuria is a risk factor in patients who do not undergo BCG treatment. Thus, because BCG treatment has been shown to have some efficacy, we believe that this treatment should be strongly considered in patients with high-risk NMIBC and pyuria. The presence of pyuria does not diminish the effectiveness of BCG treatment. BCG intravesical instillation treatment induces inflammation of the bladder, which suppresses recurrence and progression. Thus, it is contradictory to believe that its effect would be attenuated by the presence of inflammation [8,45].

The present study had some limitations. First, it was a retrospective study. Second, the study cohort was relatively small, and the study patients had been seen over a long period (2008–2022). The introduction of BCG maintenance therapy in our facility during this study period may have affected the results. A large-scale, multicenter, prospective clinical study is needed to evaluate our results.

## 5. Conclusions

To the best of our knowledge, this is the first study to evaluate the effect of preoperative pyuria on RFS according to BCG treatment status in a real-world practice setting. We found that pyuria is a risk factor for postoperative recurrence of NMIBC. BCG treatment can be expected to prevent recurrence even in patients with pyuria. Aggressive BCG treatment may need to be considered in patients with high-risk NMIBC and pyuria.

## Figures and Tables

**Figure 1 cancers-15-01638-f001:**
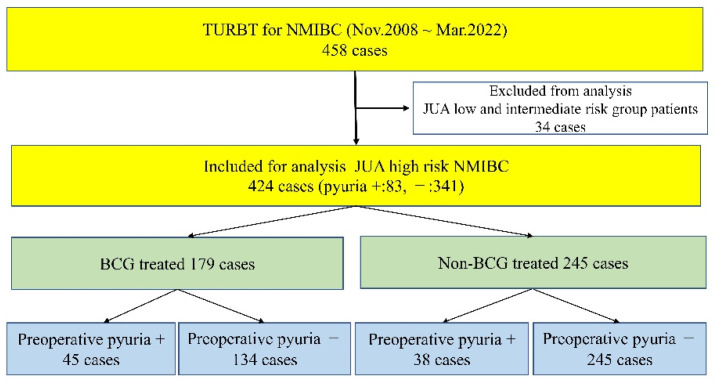
Flow chart of the inclusion and exclusion criteria and patient grouping based on risk group and presence of pyuria before trans-urethral resection of bladder tumors.

**Figure 2 cancers-15-01638-f002:**
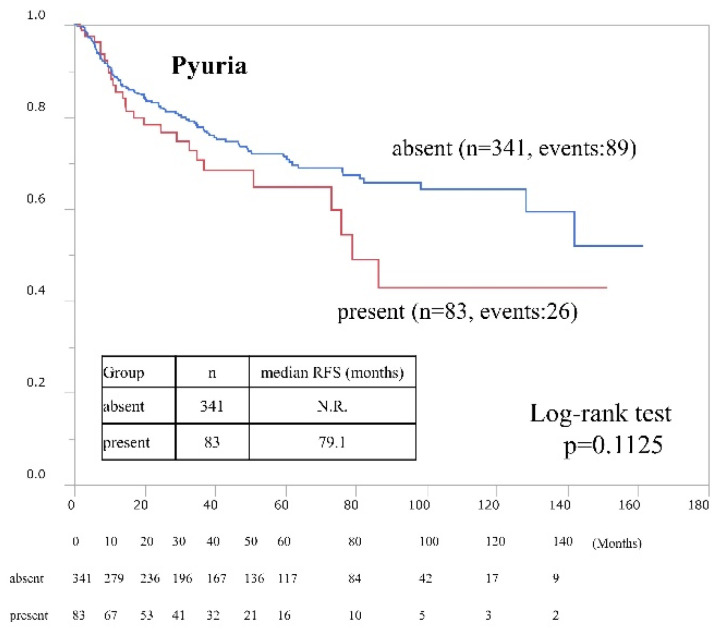
Intravesical recurrence-free survival rates after trans-urethral resection of bladder tumors according to preoperative pyuria status. Rates were estimated using the Kaplan–Meier method.

**Figure 3 cancers-15-01638-f003:**
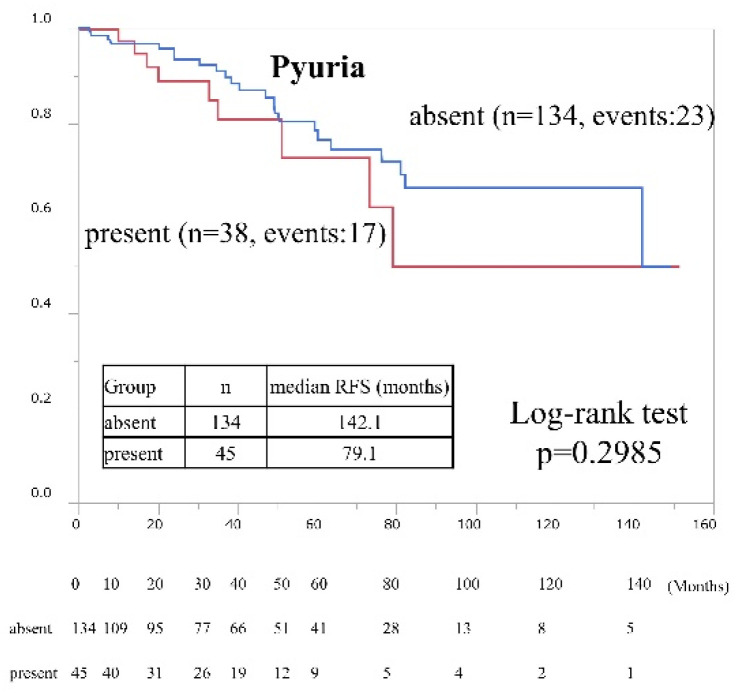
Intravesical recurrence-free survival rates after trans-urethral resection of bladder tumors in BCG treated patients according to preoperative pyuria status. Rates were estimated using the Kaplan–Meier method.

**Figure 4 cancers-15-01638-f004:**
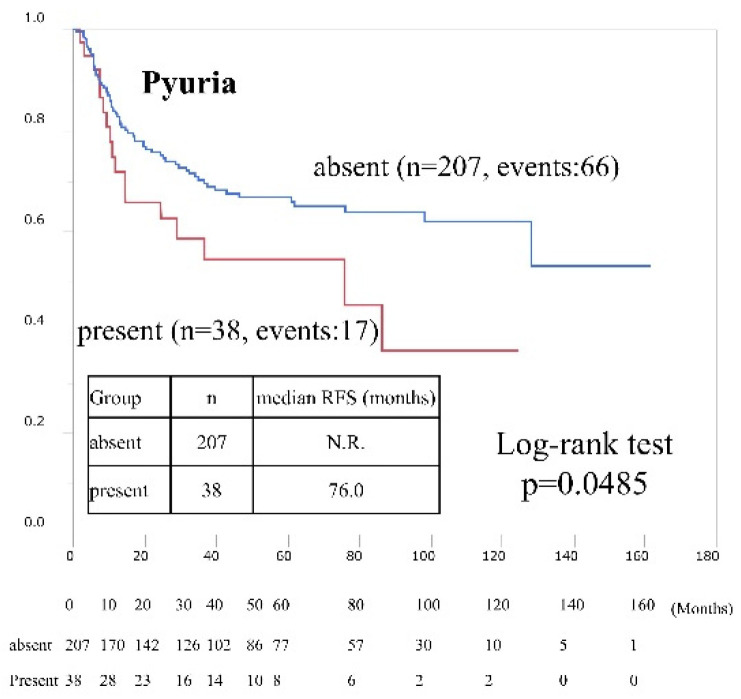
Intravesical recurrence-free survival rates after trans-urethral resection of bladder tumors in non-BCG-treated patients according to preoperative pyuria status. Rates were estimated using the Kaplan–Meier method.

**Figure 5 cancers-15-01638-f005:**
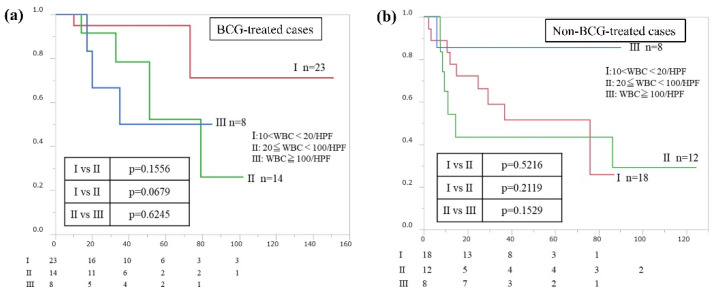
(**a**) Intravesical recurrence-free survival rates after trans-urethral resection of bladder tumors in BCG-treated patients with preoperative pyuria according to the degree of pyuria (urine WBCs 10–20/hpf, 20–100/hpf, >100/hpf). Rates were estimated using the Kaplan–Meier method. (**b**) Intravesical recurrence-free survival rates after trans-urethral resection of bladder tumors in non-BCG-treated patients with preoperative pyuria according to severity of pyuria (Urine WBCs 10–20/hpf, 20–100/hpf, >100/hpf). Rates were estimated using the Kaplan–Meier method.

**Table 1 cancers-15-01638-t001:** All patients subdivided according to the presence of preoperative pyuria.

		Total *n* = 424	Pyuria Present (*n* = 83)	Pyuria Absent (*n* = 341)	*p*-Value
Age	Median (range)	72.5 (38–94)	75 (50–90)	72 (38–94)	0.041
Sex	Male	336 (79.2)	25 (30.1)	278 (81.5)	0.019
	Female	88 (20.8)	58 (69.9)	63 (18.5)	
Preoperative cytology	Positive	83 (19.6)	19 (22.9)	64 (18.8)	0.396
	Negative	341 (80.4)	64 (77.1)	277 (81.2)	
BCG therapy	Yes	179 (42.2)	45 (54.2)	134 (39.2)	0.0136
	No	245 (57.8)	38 (46.8)	207 (60.8)	
Grade	High	325 (76.7)	69 (83.1)	256 (75.1)	0.1196
	Low	99 (23.3)	14 (16.9)	85 (24.9)	
Pathological T stage, *n* (%)	T1	109 (25.7)	32 (38.6)	77 (22.6)	0.0028
	Others	315 (74.3)	51 (61.4)	264 (77.4)	
Concomitant CIS	Present	49 (11.6)	9 (10.8)	40 (11.7)	0.827
	Absent	375 (88.4)	74 (89.2)	301 (88.3)	

**Table 2 cancers-15-01638-t002:** Results of univariate and multivariate analysis of risk factors for intravesical recurrence after TUR-Bt.

Factor	*n*	Univariate Model		Multivariate Model	
		HR (95% CI)	*p*-Value	HR (95% CI)	*p*-Value
Sex					
Male	336	1.18 (0.75–1.88)	0.4857		
Female	88				
Preoperative cytology					
Pogitive	83	1.03 (0.65–1.62)	0.9144		
Negative	341				
BCG therapy					
Yes	179	0.54 (0.36–0.81)	0.0027	0.52 (0.34–0.80)	0.0026
No	245				
Grade					
High	325	1.32 (0.87–2.01)	0.1945	0.88 (0.57–1.37)	0.5814
Low	99				
Pathological T stage					
pT1	109	0.89 (0.59–1.35)	0.5752		
Others	315				
Concomitant CIS					
Present	49	0.85 (0.47–1.56)	0.5914		
Absent	375				
Pyuria					
Present	83	1.42 (0.92–2.21)	0.1125	1.93 (1.26–2.95)	0.0329
Absent	341				

**Table 3 cancers-15-01638-t003:** Results of univariate and multivariate analysis of risk factors for RFS in BCG treated patients.

Factor	*n*	Univariate Model		Multivariate Model	
		HR (95% CI)	*p*-Value	HR (95% CI)	*p*-Value
Sex					
Male	141	1.14 (0.47–2.77)	0.7774		
Female	38				
Preoperative cytology					
Pogitive	55	0.81 (0.38–1.74)	0.5967		
Negative	124				
Grade					
High	160	0.47 (0.14–1.58)	0.2533		
Low	19				
Pathological T stage					
pT1	64	0.89 (0.59–1.35)	0.4008		
Others	115				
Concomitant CIS					
Present	32	1.59 (0.72–3.55)	0.2593		
Absent	147				
Pyuria					
Present	45	1.50 (0.69–3.27)	0.2985		
Absent	134				

**Table 4 cancers-15-01638-t004:** Results of univariate and multivariate analysis of risk factors for RFS in non-BCG treated patients.

Factor	*n*	Univariate Model		Multivariate Model	
		HR (95% CI)	*p*-Value	HR (95% CI)	*p*-Value
Sex					
Male	195	1.22 (0.71–2.11)	0.4699		
Female	50				
Preoperative cytology					
Positive	217	1.75 (0.97–3.17)	0.0645	1.84 (1.01–3.33)	0.0459
Negative	28				
Grade					
High	165	1.00 (0.63–1.59)	0.9999		
Low	80				
Pathological T stage					
pT1	45	0.87 (0.49–1.55)	0.6477		
Others	200				
Concomitant CIS					
Present	15	0.62 (0.23–1.69)	0.3567		
Absent	230				
Pyuria					
Present	38	1.70 (1.00–2.91)	0.0485	1.77 (1.03–3.04)	0.0371
Absent	207				

## Data Availability

The data presented in this study are available on request from the corresponding author.

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
