# Peer review of "Relationship between Preoperative Pyuria and Bacille Calmette-Guerin Treatment in Intravesical Recurrence after Transurethral Resection of High-Risk, Non-Muscle Invasive, Bladder Carcinoma: A Retrospective Study of Human Data"

_cancers, 2023, doi:10.3390/cancers15061638_

Round 1
Reviewer 1 Report
This is an interesting work submitted to Cancers which do clearly add significance to the field of molecular and microbiological understanding for the recurrence process over patients with BCa and undergone primary interventional treatment followed by BCG administration.
Nevertheless, the article in its current form appears with some methodological flaws which should be addressed before further evaluation.
First of all, the introduction section should be edited to introduce bias related to BCG therapy worldwide. On this purpose it would be advisable to mention and reference the existence of different BCG strains and their potential effect on survival outcomes (DOI: 10.1016/j.urolonc.2022.05.016; DOI: 10.3390/diagnostics12030586).
Furthermore, the paper is presenting too many information. For example, Table 2 and 3 are redundant. I would recommend deleting these and improving the clarity of the whole manuscript.
In conclusion, while the article address its message to the molecular BCG and Urothelial community, this should also be translated in clinical practice advice or recommendation.Finally, BCa and pyuria may be related in male population of previous chronic pelvic inflammation and conditions. I would therefore kindly ask you to refer regarding possible previous therapies such as 5-ARI or a-blockers for BPH in men which may have altered the impact of your results (DOI: 10.1111/bju.12917).It is really important to enclose track record of their changes in the updated document so as to help me in the reviewing process.I look forward to re-assess the edited version of this manuscript.
Author Response
Response to reviewer’s comments
Reviewer1
This is an interesting work submitted to Cancers which do clearly add significance to the field of molecular and microbiological understanding for the recurrence process over patients with BCa and undergone primary interventional treatment followed by BCG administration.
Nevertheless, the article in its current form appears with some methodological flaws which should be addressed before further evaluation.
First of all, the introduction section should be edited to introduce bias related to BCG therapy worldwide. On this purpose it would be advisable to mention and reference the existence of different BCG strains and their potential effect on survival outcomes (DOI: 10.1016/j.urolonc.2022.05.016; DOI: 10.3390/diagnostics12030586).
Response
I would like to express my sincere gratitude for pointing out that this is clinically very important and is the basis for this research. We add the following to the "introduction".
“BCG therapy is widely used worldwide for NMIBC patients. There are also slight differences in indications for each guideline. And previous report that was analyzed 1,228 patients with non-muscle invasive bladder cancer showed multifocality, lymphovascular invasion, and high-grade on re-TURB were independent predictors for response to BCG treatment. BCG-unresponsive patients reported worse oncological outcomes. {Ferro, 2022 #49} As mentioned above, it is true that there are some factors that are less effective for BCG treatment. There is also a report that inflammatory markers also predict the effect of BCG treatment. The modified Glasgow prognostic score has attracted attention in recent years. That might have the potential to predict recurrence in 1382 high grade/ Grade3 NMIBC patients. {Ferro, 2022 #50} In this way, various studies on BCG are currently underway, and there are many issues that must be resolved.”
Furthermore, the paper is presenting too many information. For example, Table 2 and 3 are redundant. I would recommend deleting these and improving the clarity of the whole manuscript.
Response
Thank you for your constructive feedback. Having many tables may confuse the reader.
Regarding Table 2, I think it is an important table that represents the fundamentals in discussing pyuria and BCG treatment in this study. This table confirms that BCG treatment and pyuria are independent prognostic factors in high-risk NMIBC treatment, leading to the next detailed analysis. Reviewers may perceive Tables 3 and 5 as excessive information. We will transfer this information to the supplement. (Table3→Table S1, Table5→Table S2)
In conclusion, while the article address its message to the molecular BCG and Urothelial community, this should also be translated in clinical practice advice or recommendation.Finally, BCa and pyuria may be related in male population of previous chronic pelvic inflammation and conditions. I would therefore kindly ask you to refer regarding possible previous therapies such as 5-ARI or a-blockers for BPH in men which may have altered the impact of your results (DOI: 10.1111/bju.12917).It is really important to enclose track record of their changes in the updated document so as to help me in the reviewing process.I look forward to re-assess the edited version of this manuscript.
Response
Thank you for your very kind and constructive comments. Based on the reviewer's comments, we believe that LUTs treatment will reduce the recurrence rate of NMIBC in the future. We understand that the results of this research are also important results that lead to that goal. We add the following to the "discussion".
“Based on previous studies and the results of this study, we believe that treatment for lower urinary tract symptoms in men may reduce the risk of NMIBC recurrence. Oral treatment such as alpha-blockers and 5-alpha-reductase inhibition, as well as surgical treatment, are expected to improve dysuria, thereby reducing residual urine and improving chronic urinary tract infections. {Busetto, 2015 #52} In the future, our goal is to show that lower urinary tract obstruction treatment before TUR-Bt leads to a lower recurrence rate of NMIBC.”

Reviewer 2 Report
The authors retrospectively examined real-world cases in a single institution of risk of preoperative pyuria associated with intravesical recurrence according to BCG treatment status. They showed that preoperative pyuria is not associated with intravesical recurrence after TUR-Bt in high-risk BCG-treated patients with NMIBC, but is a significant risk factor in non-BCG-treated patients. The aim is worthy of interest but requires revision.
- You classified risk according to the JUA guideline. What about Europeans and Americans? What are the differences?
- Provide a better-quality image for figure 1
- References are needed in these sentences "When treating NMIBC, it is important to suppress progression and recurrence. In particular, the intravesical recurrence rate is not low. When recurrence occurs, it creates a physical and economic burden for the patient". "The recurrence rate of NMIBC is not low and recurrences are characteristically invasive and have 49 adverse economic consequences. Suppression of recurrence requires appropriately combining perioperative intravesical instillation therapy and TUR of the bladder tumor (TURbt)".
- about reporting your previous studies I recommend including them in your discussion. this is because the introduction should be more brief and focused.
- When talking about risk factors for recurrence please include this recent paper on the topic where the diet and the meat intake have been proposed as risk factors for recurrence and for more aggressive BC.
- Since intravesical immunotherapy with BCG is the standard therapy for NMIBC a worthy focus should be done on alternative treatments such as device-assisted chemotherapy instillations (microwave-induced hyperthermia [RITE] or electromotive drug administration [EMDA]) that are experimental less effective, and significantly more expensive, further highlighting the issues of BCG shortages. A novel paper focused on survival outcomes comparing these techniques concluding that TICE was superior to RIVM for RFS outcomes (DOI: 10.3390/cancers14040887). I strongly suggest including this report in your paper as will add value to the entire manuscript.
-check typos.
Author Response
Response to reviewer’s comments
Reviewer2
The authors retrospectively examined real-world cases in a single institution of risk of preoperative pyuria associated with intravesical recurrence according to BCG treatment status. They showed that preoperative pyuria is not associated with intravesical recurrence after TUR-Bt in high-risk BCG-treated patients with NMIBC, but is a significant risk factor in non-BCG-treated patients. The aim is worthy of interest but requires revision.
- You classified risk according to the JUA guideline. What about Europeans and Americans? What are the differences?
Response
Thank you for your appropriate comment. This paper also has a lot of influence on the position of BCG in NMIBC treatment. As pointed out by the reviewers, each well-known guideline and the JUA guideline are added in the text. We have added the following to the "discussion".
Each guideline now defines risk and determines additional treatment accordingly. The representative classifications and treatment methods are shown below by risk. Currently, the highest risk is also defined, but here we show low, intermediate, and high risk, respectively. By the AUA/SUO risk stratification, low-risk NMIBC includes papillary urothelial neoplasm of low malignant potential and low-grade urothelial carcinoma that is a solitary Ta and <3 cm. a single instillation of intravesical chemotherapy immediately post-TURBT can be helpful in reducing the risk of recurrence. Intermediate-risk NMIBC includes low-grade urothelial carcinoma that has any of the following characteristics: T1, size >3 cm, multifocal, or recurrence within 1 year. In addition, high-grade urothelial carcinoma that is solitary, Ta, and <3 cm is also considered intermediate risk. After TURBT and immediate intravesical chemotherapy, the panel recommends a 6-week induction course of intravesical therapy. Options for intravesical therapy for intermediate-risk NMIBC include BCG or chemotherapy. High-risk NMIBC includes high-grade urothelial carcinoma that has any of the following characteristics: CIS, T1, size >3 cm, or multifocal. Treatment options for high-risk NMIBC depend on whether the tumor has previously been shown to be unresponsive or intolerant to BCG. For BCG-naïve NMIBC, the options are cystectomy or BCG. {Flaig, 2022 #58} By the EAU risk stratification, low-risk NMIBC includes a primary, single, Ta/T1 LG/G1 tumor <3 cm in diameter without CIS in a patient aged <70 years. “Offer one immediate instillation of intravesical chemotherapy after TURBt.” is strong recommended. Intermediate-risk NMIBC includes patients without CIS who are not included in either the low, high, or very high-risk group. “For all patients, either 1 year full-dose BCG treatment (induction plus 3-weekly instillations at 3, 6, and 12 month) or instillations of chemotherapy (the optimal schedule is not known) for a maximum of 1 year is recommended. The final choice should reflect the individual patient’s risk of recurrence and progression as well as the efficacy and side effects of each treatment modality. Offer one immediate chemotherapy instillation to patients with small papillary recurrences detected more than 1 year after previous TURBt” is strong recommended. High-risk NMIBC includes all T1 HG/G3 without CIS except those included in the very high-risk group, all CIS patients except those included in the very high-risk group, Ta LG/G2 or T1 G1 with CIS and all 3 risk factor, Ta HG/G3 or T1 LG with no CIS and at least 2 risk factor and T1 G2 with no CIS and at least 1 risk factor. (Additional risk factors: age >70 years, multiple papillary tumors, and tumor diameter >3 cm.) “Offer intravesical full-dose BCG instillations for 1–3 year or RC.” is strong recommended. {Babjuk, 2022 #59} By the JUA risk stratification, low-risk NMIBC includes initial diagnosis, <3 cm, Ta, low grade, without CIS. Low-risk patients receive single immediate intravesical instillation of anthracyclines or mitomycin C to prevent recurrence. Intermediate-risk NMIBC includes the group meets other than low and high risk. Administration of maintenance intravesical chemotherapy in addition to single immediate intravesical instillation of chemotherapy has been recommended for patients with intermediate-risk disease. but no conclusions about the regimen have been reached. In addition, intravesical BCG therapy is administered to intermediate risk. High-risk NMIBC includes the group contains any of the following factors: T1, high grade, CIS (including concurrent CIS). Intravesical BCG therapy is administered to high-risk patients. The addition of maintenance therapy after six to eight courses of intravesical induction therapy is recommended. {Matsumoto, 2020 #30}
- Provide a better-quality image for figure 1
Response
Thank you for your appropriate comment. We changed to 1200dpi Fig1.
- References are needed in these sentences "When treating NMIBC, it is important to suppress progression and recurrence. In particular, the intravesical recurrence rate is not low. When recurrence occurs, it creates a physical and economic burden for the patient".
"The recurrence rate of NMIBC is not low and recurrences are characteristically invasive and have 49 adverse economic consequences. Suppression of recurrence requires appropriately combining perioperative intravesical instillation therapy and TUR of the bladder tumor (TURbt)".
Response
Thank you for your appropriate comment. We added literature on recurrence rates and costs. We also added a review article on perioperative intravesical instillation therapy.
- about reporting your previous studies I recommend including them in your discussion. this is because the introduction should be more brief and focused.
Response
Thank you for your very kind and constructive comments. Based on the reviewer's comments, we believe that LUTs treatment will reduce the recurrence rate of NMIBC in the future. We understand that the results of this research are also important results that lead to that goal. We add the following to the "discussion".
“Based on previous studies and the results of this study, we believe that treatment for lower urinary tract symptoms in men may reduce the risk of NMIBC recurrence. Oral treatment such as alpha-blockers and 5-alpha-reductase inhibition, as well as surgical treatment, are expected to improve dysuria, thereby reducing residual urine and improving chronic urinary tract infections. {Busetto, 2015 #52} In the future, our goal is to show that lower urinary tract obstruction treatment before TUR-Bt leads to a lower recurrence rate of NMIBC.”
- When talking about risk factors for recurrence please include this recent paper on the topic where the diet and the meat intake have been proposed as risk factors for recurrence and for more aggressive BC.
Response
Thank you for your very kind and constructive comments. Thank you for the very important latest information. We add the following to the "discussion".
“And Recent study results suggest an association between bladder cancer incidence and several food items including meat, fruit, vegetables, milk products and oil. Micronutrient deficiency is associated with bladder cancer risk. It is necessary to further examine the effect of such eating habits on relapse in the future.”
- Since intravesical immunotherapy with BCG is the standard therapy for NMIBC a worthy focus should be done on alternative treatments such as device-assisted chemotherapy instillations (microwave-induced hyperthermia [RITE] or electromotive drug administration [EMDA]) that are experimental less effective, and significantly more expensive, further highlighting the issues of BCG shortages. A novel paper focused on survival outcomes comparing these techniques concluding that TICE was superior to RIVM for RFS outcomes (DOI: 10.3390/cancers14040887). I strongly suggest including this report in your paper as will add value to the entire manuscript.
Response
Thank you for your very kind and constructive comments. As the reviewers have pointed out, there are multiple types of BCG, and the effects of each type are still unclear.
The papers presented in such circumstances are very valuable, and we need to discuss them in our papers. Added the following to "discussion".
“The superiority of any BCG strain over another could not be demonstrated yet. Previous report showed that when routinely performing re-TUR followed by a maintenance BCG schedule, TICE was superior to RIVM for RFS outcomes. Further studies addressing the type of BCG may also be needed in the future.”
-check typos.
Response
Thank you for pointing this out. Before submitting this paper, we paid a native speaker to correct the English, which was indicated in the "acknowlege". I checked the text again before submitting this revised paper.

Round 2
Reviewer 1 Report
Authors answered all comments and suggestions.
Reviewer 2 Report
the revised version is worthy of publication